# Proximate Composition and In Vitro Bioactive Properties of Leaf Extracts from Seven *Viola* Species

**DOI:** 10.3390/foods14020302

**Published:** 2025-01-17

**Authors:** Guangnian Zeng, Xingfan Li, Chunbo Zhao, Yongkang Pang, Xiongfei Luo, Zhonghua Tang

**Affiliations:** 1College of Chemistry, Chemical Engineering and Resource Utilization, Northeast Forestry University, Harbin 150040, China; zengguangnian@163.com (G.Z.); zhaochunbo1992@nefu.edu.cn (C.Z.); 2023110921@nefu.edu.cn (Y.P.); 2Key Laboratory of Forest Plant Ecology, Ministry of Education, Northeast Forestry University, Harbin 150040, China; lxf1756185206@163.com

**Keywords:** wild edible plant, *Viola*, proximate composition, phenolic compounds, coumarins, bioactive properties

## Abstract

*Viola*, an edible wild plant, is valued for its distinctive flavor and health-promoting properties. This study examines the proximate composition, bioactive compounds, and in vitro biological activities of seven *Viola* leaves (*Viola prionantha*, *Viola collina*, *Viola acuminata*, *Viola variegata*, *Viola tokubuchiana* var. *takedana*, *Viola mirabilis,* and *Viola philippica*). Findings reveal that the leaves of the seven *Viola* species are rich in phenolic compounds (131.13 mg gallic acid equivalents (GAE)/100 g fresh weight (FW)–384.80 mg GAE/100 g FW), flavonoids (13.09 mg rutin equivalents (RE)/100 g FW–40.08 mg RE/100 g FW), fatty acids (palmitic acid, linoleic acid, oleic acid, and α-linolenic acid), and essential minerals, such as potassium, calcium, and magnesium. The leaf extracts demonstrated significant inhibitory effects on α-glucosidase (84.17%) and pancreatic lipase (77.54%) at a concentration of 1 g of extract per milliliter of solution. Additionally, the biological activity of *Viola* leaves, particularly their antioxidant capacity, is associated with their phenolic and flavonoid content, with caffeic acid contributing up to 35.2% of the total phenolic acids and isoquercitrin being one of the most bioavailable flavonoids. These results indicate that *Viola* offers potential notable health benefits, presenting a valuable addition to enhancing modern dietary patterns and overall health.

## 1. Introduction

Unhealthy diets and sedentary lifestyles have emerged as predominant health risks in recent decades, contributing significantly to the rising prevalence of non-communicable diseases, such as Alzheimer’s, hypertension, hyperlipidemia, and obesity. Consequently, there is an increasing need to adopt a sustainable, green, and health-oriented dietary structure [1,2]. In response to the growing demand for healthier, environmentally conscious living, wild edible plants have gained popularity in the culinary industry across various European countries and regions. These plants, integral to traditional diets, are nutrient-dense and supply high-quality fatty acids and essential trace elements, presenting a natural approach to address modern dietary deficiencies [3].

Research underscores that wild edible plants are generally abundant in vitamins, minerals, and bioactive compounds, making them valuable sources of nutrition, antioxidants, and enzyme inhibitors [4]. Among these, the genus *Viola* holds both nutritional and economic significance. As the largest genus within the *Violaceae* family, *Viola* comprises perennial or biennial herbaceous species, with over 500 identified worldwide, predominantly distributed across the Northern Hemisphere’s temperate regions, as well as Hawaii, Oceania, and the Andes of South America [5]. Notably, *Viola* contains diverse bioactive compounds, such as phenols, flavonoids, alkaloids, coumarins, and phenylpropanoids, supporting a wide range of biological functions [6]. In vitro studies confirm the extensive biological activities of *Viola* (including *Viola yedoensis*, *Viola tricolor*, *Viola odorata*, and others) extracts derived from both the whole plant and its aerial parts, obtained using water or ethanol as solvents, including antibacterial, anti-inflammatory, and antioxidant effects, underscoring their potential in disease prevention and health maintenance [7]. For example, studies utilizing a Caenorhabditis elegans obesity model revealed that *Viola tricolor* flowers exert lipid-lowering and anti-aging effects [8]. Additionally, *Viola philippica* extracts exhibited significant inhibition of Hepatitis C Virus protease and α-glucosidase and demonstrated protective effects against immunological liver damage, suggesting applications in liver-related disease management [9,10]. Similarly, *Viola odorata* L. extracts have shown efficacy in reducing blood pressure and lipid levels and supporting weight management, likely due to their antioxidant activity and inhibition of lipid synthesis [11]. Further studies reveal that *Viola odorata* L. is a potent source of antioxidants with neuroprotective effects, indicating potential therapeutic applications for neurodegenerative diseases [12]. To date, research has not reported any significant cytotoxicity associated with *Viola*, reinforcing its value as a safe, functional, plant-based resource [13].

In the Liaodong Peninsula of northeastern China, wild *Viola* leaves are a common dietary component, valued by local residents for both their nutritional and medicinal properties, including anti-inflammatory and swelling-reduction effects. Traditional records across Asia describe decoctions of dried whole plant of *Viola* as effective in disease treatment. Market surveys and sample analyses of wild *Viola* varieties popular in local markets have been conducted, focusing on proximate composition, mineral content, phenolic and flavonoid profiles, and key bioactive components. This study aims to characterize these species’ biological activities and identify bioactive compounds with nutritional and healthcare potential, maximizing their application in these fields. The properties of wild edible plants are subject to multiple influencing factors, necessitating thorough evaluation of their nutritional profile, antioxidant capacity, enzyme inhibition potential, and antibacterial activities [14,15]. In this context, a preliminary comparative analysis investigates the *Viola* genus as a viable source of nutraceuticals, antioxidants, and enzyme inhibitors. Results from this research may enhance the development and utilization of *Viola* resources, providing insights that contribute to improved dietary practices and support broader health outcomes.

## 2. Materials and Methods

### 2.1. Plant Materials

The plant materials were collected from three locations in the Liaodong Peninsula region of China (with the central coordinates being 123.232799° E, 40.287175° N, and an average collection altitude of 220 m); the *Viola* samples were identified by Professor Junlin Yu. The samples were gathered in mid-May 2023 from fully natural growth forest areas; the collection sites of wild plants are shown in Figure 1. The seven edible *Viola* species were collected as whole plants: *Viola prionantha*, *Viola collina*, *Viola acuminata*, *Viola variegata*, *Viola tokubuchiana* var. *takedana*, *Viola mirabilis*, *Viola philippica* (Figure 2a,b). The selected species were chosen based on their actual consumption by local residents and their presence in local markets in northeastern China. The specific geographical information of the collection sites is as follows: *Viola prionantha*, *Viola tokubuchiana* var. *takedana*, and *Viola acuminata* were collected from Zhuanghe City, Dalian, Liaoning Province (122.861440° E, 40.023382° N, 216 m a.s.l.). *Viola collina*, *Viola variegata*, and *Viola mirabilis* were collected from Fengcheng City, Dandong, Liaoning Province (123.814077° E, 40.742570° N, 249 m a.s.l.). *Viola philippica* was collected from Xiuyan County, Anshan, Liaoning Province (123.022881° E, 40.095574° N, 201 m a.s.l.). The fresh *Viola* samples were cleaned, flash-frozen, and stored at −20 °C, while air-dried samples were oven-dried at 60 °C, ground to powder, and stored in sealed containers for analysis.

### 2.2. Sample Processing

*Viola* leaf samples were processed following the method established by Justine Chervin et al. [16]. Fresh *Viola* plants were promptly preserved, transported to the laboratory, and processed by removing damaged parts and foreign materials before collecting the leaves. The leaves were flash-frozen in liquid nitrogen and stored at −20 °C. To obtain dry plant samples, the fresh samples were first air-dried and then placed in an oven at 60 °C for over 48 h until a constant weight was achieved. The dried samples were ground into powder using a grinder, sieved through a 20-mesh screen, and stored in a sealed container in a cool, ventilated, dry place. The samples were extracted with 80% ethanol (EtOH:H_2_O, 80:20 *v*/*v*) by adding 10 mL of the solution to 0.50 g of fresh leaf samples. After ultrasonic treatment at room temperature and centrifugation, the supernatant was collected. This extraction process was repeated, and the two supernatants were combined to create a crude ethanol extract of *Viola* leaves. The combined supernatants were used for nutrient, physicochemical, and bioactivity assays, while mineral and fatty acid analyses were performed on dried samples.

### 2.3. Proximate Composition

#### 2.3.1. Soluble Sugars and Proteins

The anthrone-sulfuric acid colorimetric method quantified soluble sugar content: 2 mL of anthrone-sulfuric acid solution (0.03 g/mL) was added to the crude ethanol extract, incubated in boiling water for 10 min, then cooled to room temperature, with absorbance recorded at 610 nm on a UV spectrophotometer. Results were expressed as grams per 100 g of fresh weight. Protein content was measured using the Kjeldahl method, with samples (0.20 g) digested using Kjeldahl catalyst tablets and sulfuric acid [17].

#### 2.3.2. Total Phenols and Total Flavonoids

For total phenolic and flavonoid analyses, extracts were diluted with 80% ethanol. The Folin–Ciocalteu method determined phenolic content, measured at 765 nm and expressed as mg of gallic acid equivalents (GAE) per 100 g of extract [18]. Flavonoid content was assessed following the method by Guo et al., with absorbance at 415 nm and expressed as mg of rutin equivalents (RE) per 100 g of extract [19].

#### 2.3.3. Total Carotenoids, Chlorophyll, and Ascorbic Acid

To measure total carotenoid and chlorophyll content of the seven *Viola* leaves, 0.1 g samples were soaked in 10 mL of 80% acetone for 24 h, centrifuged, and absorbance recorded at 663, 645, and 470 nm. Ascorbic acid content was quantified using UHPLC with an Agilent Eclipse Plus C18 column (Agilent Technologies, Santa Clara, CA, USA) and detection at 245 nm, referencing a standard curve.

#### 2.3.4. Fatty Acids

Medium- and long-chain fatty acids were analyzed through ion/selected ion monitoring, with modifications based on Thurnhofer et al.’s methodology [20]. Thawed samples (at 4 °C) were extracted using a dichloromethane–methanol solution (2:1 *v*/*v*). Extracts were washed with deionized water, dried under nitrogen gas, redissolved in n-hexane, and methylated. Final analysis was conducted by gas chromatography–mass spectrometry (GC-MS) using an Agilent 19091S-433UI capillary column (Agilent Technologies., Santa Clara, CA, USA) and helium as the carrier gas. Mass spectrometric analysis utilized an Agilent 5977B MSD (Agilent Technologies, Santa Clara, CA, USA) with electron impact (EI) ionization in SCAN/SIM mode. Data, including standard curve construction and fatty acid content calculations, were processed using MSD ChemStation software (G1701DA Rev. D.01.00–SP 1).

#### 2.3.5. Mineral Composition

Mineral content in *Viola* leaves was analyzed using an Inductively Coupled Plasma-Optical Emission Spectroscopy (ICP-OES) device (PerkinElmer, Optima 8300, Waltham, MA, USA). The process involved digesting dehydrated sample powder with a 5:1 mixture of nitric and perchloric acids. Following established protocols, the elemental content, specifically aluminum (Al), calcium (Ca), copper (Cu), iron (Fe), potassium (K), magnesium (Mg), manganese (Mn), molybdenum (Mo), zinc (Zn), and sodium (Na), was determined by referencing the calibration curve from standard solutions, blank samples, and test samples, with background absorption subtracted for accurate values.

#### 2.3.6. Untargeted Primary Metabolites Profiling of Plant Samples Using GC-MS

Plant samples were extracted following the procedure outlined in Section 2.2, with modifications based on Li et al. [21]. In brief, 0.9 g of the sample was combined with cold methanol, L-2-chlorophenylalanine, and chloroform, followed by ultrasonication and centrifugation. The resulting residue was derivatized and analyzed using an Agilent 7890-5977B GC-MS system (Agilent Technologies, Santa Clara, CA, USA) equipped with an HP-5 ms Ultra Inert 7-inch Capillary Column (30 m × 0.25 mm × 0.25 μm, temperature limits: −60 °C to 325 °C). High-purity helium gas (He ≥ 99.999%) was supplied by Harbin Qinghua Gas Industry (Harbin, China). The system was equipped with an Agilent 7693A automatic liquid sampler (Product Number: G4513A).

#### 2.3.7. Characterization of Phenolic and Flavonoid Compounds by LC-MS

A targeted quantitative analysis of 34 phenolic and flavonoid compounds was conducted based on previous research methods from the research group [22,23]. Briefly, a fresh plant sample (0.15 g) was mixed with 10 mL of 70% methanol (methanol: H_2_O, 70:30, *v*/*v*) and subjected to ultrasonic extraction at 40 °C for 40 min. After centrifugation, the supernatant was collected for subsequent instrument analysis.

#### 2.3.8. Identification and Quantitative Analysis of Coumarins Using UHPLC

The identification and quantification of four coumarin compounds in the seven *Viola* species were performed using ultra-high-performance liquid chromatography (UHPLC). Chromatographic conditions included an Agilent Eclipse Plus C18 column (3.0 × 150 mm), with a mobile phase of 0.02% phosphoric acid aqueous solution and acetonitrile (40:60 *v*/*v*) at a flow rate of 1.0 mL/min. The detection wavelength was set at 342 nm, with a 20 µL injection volume, and a column temperature of 30 °C. A standard curve was developed using a mixed solution of the four coumarin standards at various concentrations, allowing for the measurement and calculation of coumarin content across different samples. See Appendix A for the chromatogram, the regression equation, correlation coefficient, and limit of detection of the compounds identified.

### 2.4. Bioactive Activity

#### 2.4.1. Antioxidant Activity in Vitro

The DPPH radical scavenging activity was measured following Sarmento et al.’s method, with anhydrous ethanol as the control [24]. Extracts at varying concentrations were combined with DPPH radicals and incubated in darkness for 30 min, after which absorbance was recorded at 517 nm. IC50 values were calculated using ascorbic acid as the reference standard. The ABTS radical scavenging activity and ferric ion reducing antioxidant power (FRAP) were assessed according to Sharma et al. [25]. For ABTS, a radical cation solution was prepared and diluted to reach an absorbance of 0.70 ± 0.02 at 734 nm, mixed with extracts, and absorbance was measured after 6 min. FRAP was analyzed using a standard curve with ferrous sulfate, with results expressed as mg ferrous sulfate per 100 mg of extract.

#### 2.4.2. Antimicrobial Activity

Antimicrobial activity of the seven wild *Viola* species was evaluated by determining inhibition zone diameter, minimum inhibitory concentration (MIC), and minimum bactericidal concentration (MBC) against *Staphylococcus aureus* (ATCC 25923), *Escherichia coli* (ATCC 25922), *Pseudomonas aeruginosa* (ATCC 15442), and *Candida albicans* (ATCC 10231). These strains were cultured in a Luria–Bertani medium at 37 °C to a concentration of 106 CFU/mL [26].

The inhibition zone diameter, MIC, and MBC were evaluated based on the protocols outlined by Leja and Zhao et al., with sterile water and kanamycin as controls for inhibition zone assessment and ampicillin as the positive control for MIC and MBC measurements [27,28].

#### 2.4.3. Enzyme Activity Inhibition Capacity

The inhibitory effects of *Viola* on pancreatic lipase and α-glucosidase were assessed as follows: for pancreatic lipase inhibition, 0.1 g of porcine pancreatic lipase was suspended in 5 mL of Tris-HCl buffer (50 mM, pH 7.2–7.4, containing 0.1% gum arabic and 0.2% sodium deoxycholate) and centrifuged at 2000 g for 10 min. In a 96-well plate, 10 μL of the sample, 30 μL of Tris-HCl buffer, and 150 μL of enzyme solution were combined and incubated at 37 °C for 20 min, followed by the addition of 10 μL of p-nitrophenyl palmitate (PNPP, 10 mM). Absorbance at 405 nm was monitored over 20 min. For α-glucosidase inhibition, 140 μL of PBS (pH 7.2–7.4), 20 μL of α-glucosidase (7 units/mL), and 20 μL of the sample or control were incubated at 37 °C for 20 min, after which 20 μL of p-nitrophenyl-α-D-glucopyranoside (2.5 mM) was added. Following a further 20 min incubation at 37 °C, absorbance at 405 nm was recorded to calculate enzyme inhibition.

### 2.5. Statistical Analysis

All experiments were conducted in triplicate, and data analysis involved analysis of variance (ANOVA), followed by Duncan’s post hoc test using SPSS software (version 22.0; IBM Corp., Armonk, NY, USA). Graphs were compiled using GraphPad Prism 8.0 (GraphPad Software, San Diego, CA, USA), Adobe Illustrator 2022 (Adobe Inc., San Jose, CA, USA), and ArcMap 10.8 (ArcGIS Software, Redlands, CA, USA).

## 3. Results and Discussion

### 3.1. Proximate Composition

#### 3.1.1. Physicochemical Characterization

The data on the seven *Viola* species (*Viola prionantha*, *Viola collina*, *Viola acuminata*, *Viola variegata*, *Viola tokubuchiana* var. *takedana*, *Viola mirabilis*, and *Viola philippica*) for leaf biomass, water content, soluble sugars, protein, total phenolic and flavonoid contents, carotenoids, chlorophyll, ascorbic acid, and fatty acid composition are detailed in Table 1. The dry matter ranged from 0.19 g to 1.04 g per whole *Viola* plant, with water content between 82% and 90%, reflecting significant interspecies differences in biomass. Protein content among the species showed considerable variability, with values between 1.35% FW and 5.74% FW. Notably, *Viola acuminata* and *Viola mirabilis* exhibited higher protein levels at 5.74 ± 0.17% FW and 5.49 ± 0.01% FW, respectively, which is higher than that in *Raphanus sativus* L. leaves (3.81 ± 0.32 g/100 g DW) [29]. Soluble sugar content in the seven *Viola* leaves ranged from 4.70% FW to 7.31% FW, higher than levels found in vegetables like *Brassica oleracea* and *Raphanus raphanistrum* [30]. The slightly higher soluble sugar content may be attributed to the fact that the *Viola* plants collected were in their seedling stage, selected for better flavor and texture. During early growth, leaves generally have higher soluble sugar content to support plant development. This indicates the nutritional richness of *Viola* as a potential dietary resource.

The total phenolic content in the seven *Viola* leaves ranged from 131.13 mg GAE/100 g FW to 384.80 mg GAE/100 g FW, while the total flavonoid content varied between 15.64 mg RE/100 g FW and 40.08 mg RE/100 g FW, aligning well with values reported by Kaundal et al., confirming the reliability of their findings [31]. Among the species, *Viola mirabilis*, *Viola acuminata*, and *Viola prionantha* exhibited the highest phenolic levels, at 384.80 ± 8.94 mg GAE/100 g FW, 377.41 ± 105.47 mg GAE/100 g FW, and 359.16 ± 93.46 mg GAE/100 g FW, respectively, with specific phenolic compositions detailed in Section 2.2. Although phenolic and flavonoid levels across the seven wild *Viola* species generally fell within the expected range, some variation was noted, primarily due to environmental factors such as altitude, soil conditions, and geographical location. In terms of total chlorophyll and carotenoid content, carotenoid content ranged from 4.31 mg/100 g FW to 52.21 mg/100 g FW, while chlorophyll levels varied from 101.12 mg/100 g FW to 415.52 mg/100 g FW, demonstrating interspecies differences. *Viola acuminata* displayed the highest ascorbic acid content, reaching 404.12 ± 77.31 mg/100 g FW, followed by *Viola* mirabilis (259.89 ± 55.56 mg/100 g FW) and *Viola philippica* (198.94 ± 10.31 mg/100 g FW). Known as vitamin C, ascorbic acid is a potent antioxidant beneficial to human health, with the ability to synergize with tocopherols to boost antioxidant efficacy [32].

#### 3.1.2. Fatty Acids

This study identified 40 types of fatty acids in the leaves of seven wild Viola species, quantifying total fatty acids, saturated fatty acids (SFA), polyunsaturated fatty acids (PUFA), and monounsaturated fatty acids (MUFA). Detailed fatty acid types and concentrations are shown in Table 2 and Figure 3a. A quality control check validated data reliability, with the PCA plot displayed in Appendix A. Detected fatty acids in Viola leaves included ω-3 and ω-6 PUFAs, such as 5Z,8Z,11Z,14Z,17Z-eicosapentaenoic acid (EPA), 7Z,10Z,13Z,16Z,19Z-docosapentaenoic acid (DPA), 4Z,7Z,10Z,13Z,16Z,19Z-docosahexaenoic acid (DHA), γ-linoleic acid, and arachidonic acid (ARA), with composition illustrated in PUFA 3b. Among species analyzed, *Viola philippica* had the highest total fatty acid content, followed by *Viola acuminata* and *Viola variegata*. Of the thirty-nine identified fatty acids, eight were MUFA, fifteen were PUFA, and sixteen were SFA. As shown in Figure 3b, unsaturated fatty acids were significantly more abundant than saturated fatty acids, with polyunsaturated fatty acids as the primary component, followed by saturated and, to a lesser extent, monounsaturated fatty acids. Balanced fatty acid intake is essential in human diets. While saturated fatty acids, commonly found in foods like fatty meats, butter, and lard, contribute to energy intake, unsaturated fatty acids, particularly ω-3 and ω-6 PUFAs, offer notable health benefits [33]. The World Health Organization advises reducing saturated fat intake to below 10% of total energy and replacing saturated and trans fats with unsaturated fats, especially polyunsaturated fats. Additionally, a balanced intake of ω-3 and ω-6 PUFAs is critical. Modern diets often have an ω-6 to ω-3 PUFA imbalance, which may increase cardiovascular and inflammatory disease risk [34]. The ω-3 to ω-6 PUFA ratio in *Viola* leaves ranged from 0.8 to 2.5, averaging a near-ideal 7:4 balance that optimally supports fatty acid function in the body. ω-3 PUFAs, abundant in foods like fish, flaxseed, and walnuts, are recognized for their anti-inflammatory properties. In similar studies, researchers have identified fatty acids, such as arachidonic acid, palmitoleic acid, oleic acid, linoleic acid, and α-linolenic acid, in the edible flowers of pansy (*Viola*). This finding highlights the widespread presence of these fatty acids across plants in the *Viola* genus [13]. Among the fatty acids identified in the seven *Viola* leaves, α-linolenic acid had the highest concentration, followed by palmitic acid, linoleic acid, oleic acid, and stearic acid. As an ω-3 PUFA, α-linolenic acid is highly valued for its role in maintaining cognitive function in adults, preventing neurodegenerative conditions, and reducing cardiovascular disease risk, making it an economically significant polyunsaturated fatty acid [35]. Linoleic acid, an ω-6 PUFA, and oleic acid, a MUFA, also contribute to the fatty acid profile. The presence of palmitic acid, a saturated fatty acid, corroborates findings from Section 3.1.1. Although often limited in dietary recommendations, palmitic acid plays a valuable role in energy balance. In fact, a balanced intake of saturated fatty acids, alongside unsaturated fatty acids, is essential for maintaining optimal energy levels and supporting a well-rounded diet.

#### 3.1.3. Mineral Composition

The essential minerals in the seven *Viola* species, including Mg, Ca, Mn, Zn, Na, K, Fe, Al, Mo, and Cu, are outlined in Appendix A and Figure 3c. These leaves are particularly rich in macro-elements like K, Ca, and Mg while showing a low Na content, making them a valuable dietary source. Approximately 70 g of *Viola* leaves can fulfill daily potassium requirements. Cooking, particularly boiling, can reduce mineral content by 20% to 60%; therefore, raw consumption in salads is recommended to retain their nutrient profile and enhance flavor [36].

#### 3.1.4. Identification of Primary Biometabolites in the Seven Viola Species

Using GC-MS analysis, 58 common primary metabolites were identified in ethanol extracts from the leaves of seven wild *Viola* species. These compounds included 23 organic acids, 16 sugars, 5 amino acids, and 14 additional substances. Key organic acids with high abundance were glyceric acid, malic acid, citric acid, and D-gluconic acid, while the most prominent sugars included D-talose, D-fructose, sucrose, and D-glucose. Detected amino acids comprised L-serine, L-threonine, L-aspartic acid, L-isoleucine, and L-valine, with L-isoleucine, L-threonine, and L-valine classified as essential amino acids. Additionally, two saturated fatty acids, palmitic acid and stearic acid, along with myo-inositol, important for cellular function and metabolism, were identified across all seven *Viola* species, underscoring the leaves’ high nutritional value with compounds vital for human health [37].

#### 3.1.5. Distribution of Phenolic, Flavonoid, and Coumarin Compounds of Seven Viola Species

Phenolic compounds are essential for human health, particularly in their ability to modulate digestive enzymes like lipase, which plays an essential role in dietary lipid hydrolysis. Wild edible plants, known for their phenolic richness, exhibit various biological activities associated with these compounds [24]. In this study, targeted liquid chromatography–mass spectrometry (LC-MS) was utilized to quantitatively profile the phenolic compounds in the seven *Viola* leaves, identifying 22 phenolic and flavonoid compounds as shown in Table 2 and Appendix A. Of these, 10 were phenolic acids (No. 1–No. 11), including p-hydroxycinnamic acid, benzoic acid, gentianic acid, eugenic acid, cinnamic acid, caffeic acid, procatechin, vanillic acid, rosmarinic acid, and mustelic acid. The remaining 12 were flavonoids (No. 12–No. 22), such as genistein, apigenin, naringin, quercetin, catechin, kaempferol, glycyrrhizin, galangin, rutin, genistin, echinacoside, and isoquercitrin. Predominant compounds included isoquercitrin, catechin, caffeic acid, apigenin, genistein, p-hydroxycinnamic acid, and vanillic acid, with notable interspecies variability in concentrations (Appendix A). *Viola acuminata* displayed the highest content of phenolic and flavonoid compounds, followed by *Viola mirabilis*.

The loading plot (Figure 5c) highlights relationships among the phenolic and flavonoid compounds, with distances between points indicating correlation strengths. Among the compounds in the seven *Viola* species, quercetin and catechin showed the greatest influence on the dataset, with catechin and apigenin contributing most to variance. Catechin, a potent antioxidant, was abundant in *Viola acuminata* leaves, reaching 1238.47 ± 246.33 µg/kg FW [38]. Additionally, *Viola acuminata* and *Viola mirabilis* had high genistein and apigenin content, compounds known for health benefits, genistein may support women’s health, while apigenin offers anti-tumor and anti-inflammatory benefits, contributing to cardiovascular health [39]. *Viola variegata* exhibited the highest vanillic acid content at 689.6 ± 33.73 µg/kg FW and comparable levels of isoquercitrin and other compounds. The diversity and bioactivity of these phenolic and flavonoid compounds suggest *Viola* species are promising candidates for functional food research and potential applications in health-promoting products.

*Viola* is recognized for its medicinal properties, particularly due to its characteristic bioactive coumarins [40]. Coumarins, a subclass of phenylpropanoid compounds, demonstrate notable biological activity by forming non-covalent interactions with enzymes and receptors in biological systems [41]. This study focused on four coumarins, esculetin, esculin, scopoletin, and umbelliferone, previously identified in the seven *Viola* species. UHPLC was used to analyze the distribution of these compounds across the seven wild *Viola* samples. Results revealed the absence of scopoletin and umbelliferone in the tested samples, while esculetin and esculin were present. Esculetin and esculin, known for their anti-inflammatory and antioxidant properties, have been documented to exhibit synergistic effects [42]. In the tested *Viola* species, esculin content ranged from 8282.43 ± 801.62 μg/kg to 60,446.09 ± 9115.45 μg/kg, and esculetin levels varied between 4062.15 ± 44.25 μg/kg and 10,502.51 ± 1399.32 μg/kg. Notably, this study is the first to report the specific distribution of esculetin in *Viola acuminata* and *Viola variegata* and the presence of esculin in *Viola prionantha*, providing new insights into the bioactive potential of these wild *Viola* species.

### 3.2. Bioactive Activity

#### 3.2.1. In Vitro Antioxidant Activity

Antioxidant data for ethanol extracts of the seven wild *Viola* species leaves is detailed in Appendix A, with the antioxidant capacity expressed as the concentration of extract needed to achieve a 50% radical scavenging rate (IC50). Antioxidant activity was assessed using DPPH and ABTS radical scavenging assays, along with ferric-reducing antioxidant power. Results indicated that *Viola tokubuchiana* var. *takedana* and *Viola philippica* exhibited the highest DPPH and ABTS radical scavenging activities, as evidenced by their low IC50 values. In the FRAP assay, *Viola variegata* demonstrated the strongest iron-reducing capacity (0.13 ± 0.02 mg FE/g). These variations in antioxidant capacity across *Viola* species are likely attributable to differing concentrations of active compounds, such as polyphenols and flavonoids. These results underscore the potential of *Viola* species as natural antioxidant sources, offering promise for functional health food development.

#### 3.2.2. Antimicrobial Activity

The antimicrobial activity of the seven wild *Viola* species against five microbial strains, *Staphylococcus aureus*, *Escherichia coli*, *Pseudomonas aeruginosa*, and *Candida albicans*, was evaluated using inhibition zone diameter, MIC, and MBC, as shown in Table 3 Ethanol extracts displayed inhibitory effects on microorganisms, with greater efficacy against Gram-positive bacteria compared to Gram-negative and relatively limited activity against fungi. *Viola philippica* and *Viola variegata* exhibited the strongest inhibitory effects overall. Specifically, *Viola philippica* was most effective against *Staphylococcus aureus* and *Escherichia coli*, while inhibition of *Pseudomonas aeruginosa* was relatively consistent across species. The extracts demonstrated a more pronounced antibacterial effect than antifungal, with *Viola philippica* and *Viola variegata* showing the highest activity. MIC and MBC data were consistent with the inhibition zone results, reinforcing the potential of *Viola* species as natural antimicrobial agents.

#### 3.2.3. Enzyme Activity Inhibition Capacity

To further explore the bioactivity of the seven *Viola* extracts, in vitro studies were conducted on enzymes related to diabetes and hyperlipidemia, specifically α-glucosidase and pancreatic lipase, both of which serve as markers for non-communicable diseases and are key digestive enzymes [43,44]. The high phenolic content, including specific flavonoids (e.g., quercetin and kaempferol derivatives) and other phytoconstituents identified in *Viola* leaves, has been shown in previous research to exert inhibitory effects on digestive enzymes in vitro [45]. These phytochemicals are known for their ability to bind to the active sites of α-glucosidase and pancreatic lipase, potentially altering enzyme activity and reducing substrate cleavage efficiency. Detailed enzyme inhibition data for the seven wild *Viola* extracts are presented in Appendix A and Figure 4. Figure 4a,c display the inhibition rates and IC50 values for α-glucosidase at various concentrations, while Figure 4b,d illustrate the inhibition rates and IC50 values for pancreatic lipase. The crude *Viola* extracts demonstrated significant inhibitory effects on α-glucosidase and pancreatic lipase, with IC50 values ranging from 0.35 ± 0.11 mg/mL to 2.95 ± 0.21 mg/mL for α-glucosidase and from 0.63 ± 0.03 mg/mL to 1.18 ± 0.12 mg/mL for pancreatic lipase. In a comparable study, a 50% ethanol extract of frangipani flower at 1 mg/mL exhibited approximately 90% inhibition of α-glucosidase, aligning with the inhibitory levels observed in the seven *Viola* extracts in this study, suggesting robust reliability of the results [46]. Additionally, the enzyme inhibition capacity of the extracts was positively correlated with increased concentrations; at 1.0 mg/mL, α-glucosidase activity inhibition approached 80%, while pancreatic lipase inhibition reached around 90%. Previous studies suggest that phenolic compounds inhibit α-glucosidase and pancreatic lipase through competitive binding at the enzyme’s active site or by altering its conformation through hydrogen bonding and π-π interactions [47]. Moreover, the antioxidant properties of these phytoconstituents may further stabilize the enzyme–substrate complex, enhancing inhibitory efficiency. This finding suggests that a moderate intake of *Viola* extracts may potentially reduce α-glucosidase and lipase activity, offering possible benefits for individuals with diabetes, obesity, and related conditions. Among the tested species, *Viola prionantha* (78.80%), *Viola mirabilis* (78.41%), and *Viola acuminata* (78.39%) exhibited the highest α-glucosidase inhibition rates. Significant differences in enzyme inhibition were observed across the seven *Viola* species (*p* < 0.05), with comparisons made solely among the extracts, excluding control groups. For pancreatic lipase inhibition, *Viola prionantha* (77.54%) and *Viola tokubuchiana* (77.32%) were the most effective. Remarkably, these extracts exhibited higher inhibition rates than standard drugs acarbose and orlistat, possibly due to the solid starch content in control tablets, as starch granules do not dissolve uniformly in water, potentially diminishing their inhibitory efficacy. Although these bioactivity assessments were conducted in vitro under conditions replicating physiological pH (7.2–7.4) and temperature, further in vivo investigations are warranted to substantiate and expand upon the bioactive potential of *Viola* extracts across distinct plant parts.

### 3.3. Correlation Analysis Between Chemical Composition and Bioactive Functions

The interaction heatmap (Figure 5a) illustrates the correlations between the bioactivity of ethanol crude extracts from the seven *Viola* leaves and their chemical components, using Pearson’s correlation coefficient (*p* < 0.05). The legend on the right side indicates whether these correlations are positive or negative. As shown in Figure 5a, the bioactivity of the seven *Viola* extracts is closely associated with their chemical composition. Specifically, the antibacterial activity of the extracts shows a highly significant correlation with total flavonoid content (*p* < 0.01) and a significant correlation with caffeic acid content (*p* < 0.05). Furthermore, both lipase inhibitory activity and antioxidant capacity exhibit highly significant correlations with total phenolic content, suggesting that the observed bioactivity of the seven *Viola* leaves is predominantly linked to phenolic and flavonoid compounds, which aligns with findings by Alam et al. [48]. This likely reflects the redox properties and unique chemical structures of phenolic compounds that contribute to their strong antioxidant effects [49].

Additionally, the antioxidant capacity of the extracts significantly correlates with caffeic acid and isoquercetin levels, while ascorbic acid also shows a positive correlation with both antioxidant and antibacterial activities. Caffeic acid, a critical polyphenolic compound in the phenylpropanoid family, is known for its potent antioxidant and anti-inflammatory effects through its regulation of multiple metabolic pathways [50]. The structural characteristics of caffeic acid, particularly the hydroxyl groups in the ortho-position, enable effective radical scavenging and metal ion chelation, which likely contribute to its antioxidant and antibacterial properties [51]. Flavonoids, similarly, offer substantial health benefits. Quercetin, a well-studied flavonoid, not only provides general polyphenolic benefits but also inhibits viral replication by interfering with the viral lifecycle and supports blood pressure regulation and vascular health. Isoquercetin, a glycosylated derivative of quercetin, is particularly significant. Glycosylated bioactives often have enhanced health benefits due to their limited natural presence in foods and low water solubility, which restricts bioavailability. However, isoquercetin is more readily absorbed in the gut, where it quickly converts to quercetin in vivo, providing amplified effects [52]. Phenolic and flavonoid compounds are known to disrupt bacterial membranes by altering membrane permeability and inducing leakage of intracellular contents. Additionally, these compounds can inhibit bacterial enzymes involved in DNA replication and protein synthesis, further enhancing their antibacterial effects. These mechanisms may explain the significant correlations observed between total flavonoid content and the antibacterial activity of *Viola* extracts.

Coumarin compounds are generally regarded as key bioactive components in the seven *Viola* species. In this study, two coumarins, esculetin and esculin, were identified and quantified in *Viola* extracts, although they did not show significant correlations with bioactivity in the interaction analysis. This may be due to the presence of other coumarins in effective concentrations or the predominant influence of phenolic and flavonoid compounds, resulting in potential synergistic or antagonistic effects among compounds. Further studies focusing on the purification of individual compounds and functional validation experiments are recommended to confirm these results and to evaluate the potential role of coumarin compounds in the seven *Viola* species. It should be noted that the correlations identified between bioactive components and biological activities are based on crude extracts, which may not account for potential synergistic among individual compounds. Further studies are needed to comprehensively assess this aspect.

## 4. Conclusions

This study demonstrated that the wild edible plant *Viola* is a rich source of essential nutrients, including protein, organic acids, phenolic and flavonoid compounds, as well as ω-3 fatty acids like EPA and DHA, which are beneficial for heart health. The seven *Viola* extracts displayed substantial antioxidant activity and strong enzyme inhibition, primarily linked to their phenolic and flavonoid content, especially caffeic acid and isoquercetin. These findings highlight *Viola*’s potential in enhancing dietary quality and preventing disease, suggesting applications as a natural source for vegetables, supplements, and pharmaceuticals, particularly as antioxidants and enzyme inhibitors. Future studies are encouraged to isolate and characterize individual bioactive compounds from *Viola* species to gain a deeper understanding of their specific mechanisms of action and to explore their potential applications in functional foods, nutraceuticals, and pharmaceuticals.

## Figures and Tables

**Figure 1 foods-14-00302-f001:**
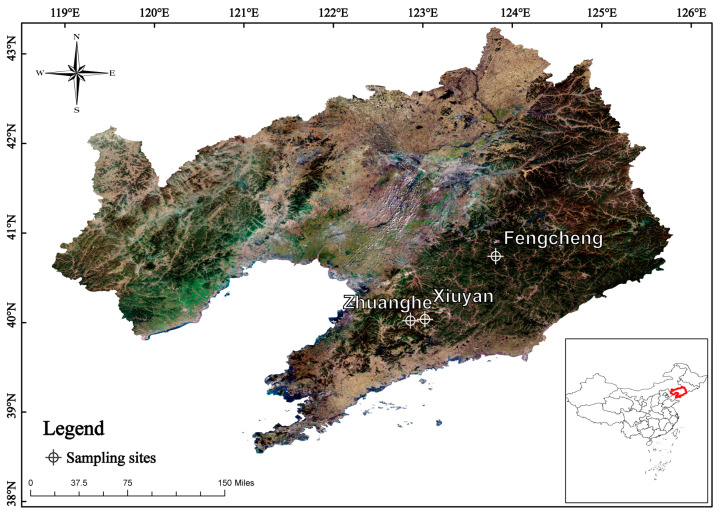
Map of the seven *Viola* species collection sites within the Liaodong Peninsula study area, Liaoning Province, China.

**Figure 2 foods-14-00302-f002:**
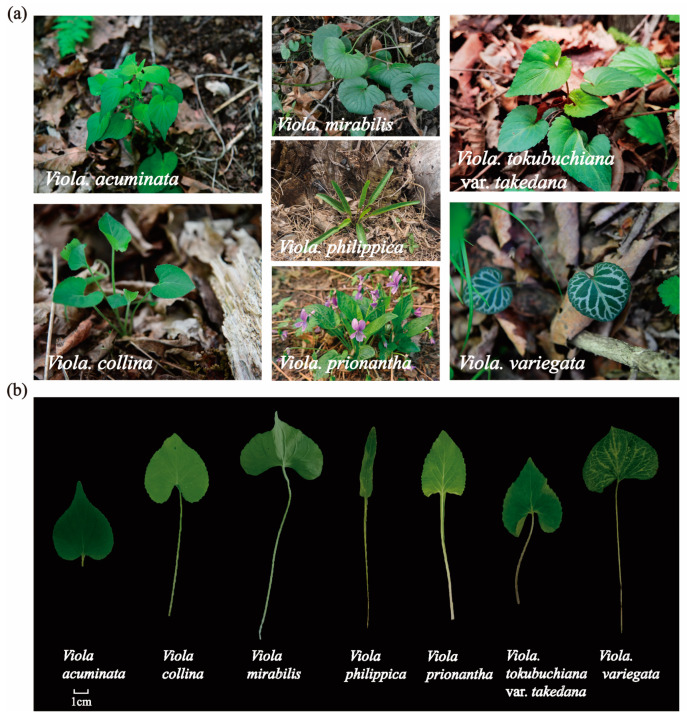
(**a**) Images of the environment of the seven *Viola* species collected. (**b**) Leaves of the seven *Viola* species.

**Figure 3 foods-14-00302-f003:**
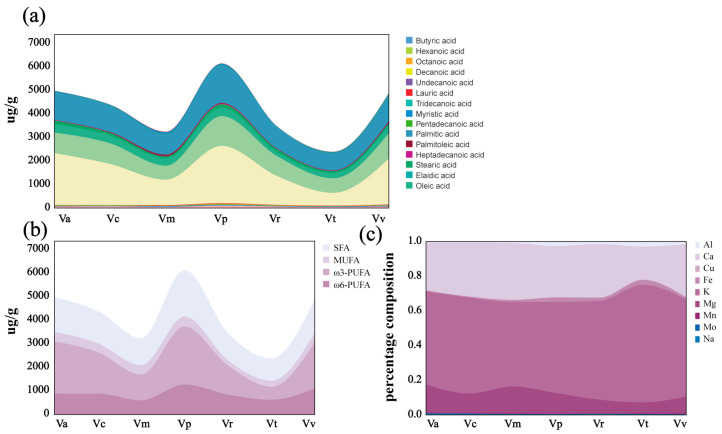
The composition of fatty acids and mineral ions in the leaves of the seven *Viola* species. (**a**) Distribution of individual fatty acids across the species. (**b**) Breakdown of fatty acids by type, showing saturated fatty acids (SFA), monounsaturated fatty acids (MUFA), polyunsaturated fatty acids (PUFA), and the specific categories of ω-3 and ω-6 polyunsaturated fatty acids present in the seven *Viola* leaves. (**c**) Composition of mineral ions in the seven *Viola* leaves.

**Figure 4 foods-14-00302-f004:**
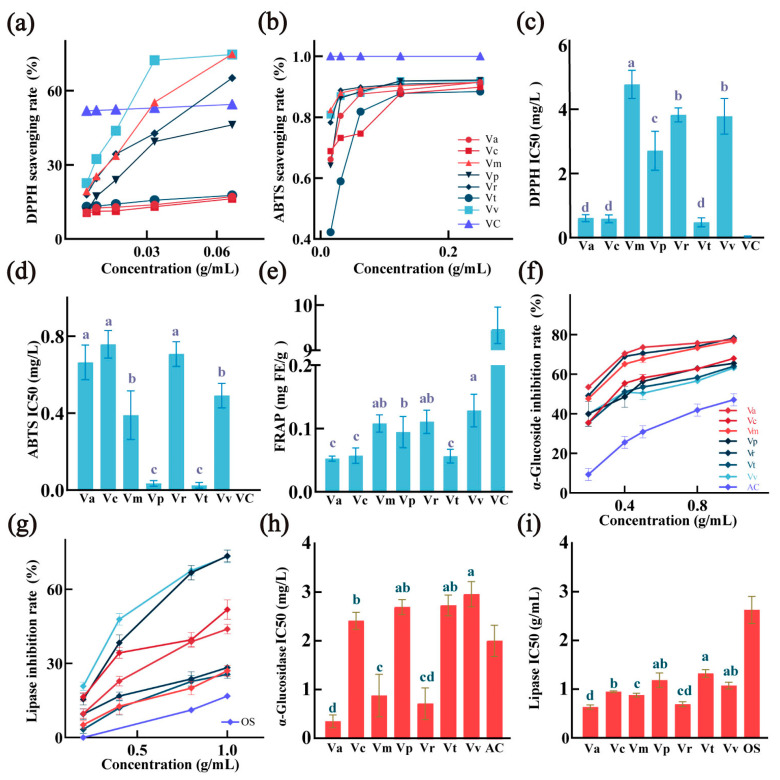
The antioxidant and enzyme inhibition activities of the seven *Viola* leaves: (**a**) DPPH scavenging rate across various concentrations, the legend’s meaning is consistent with that in (**b**). (**b**) ABTS scavenging rate. (**c**) IC50 values for DPPH scavenging. (**d**) ABTS scavenging activity. (**e**) FRAP assay results. (**f**) α-glucosidase inhibition rate. (**g**) Lipase inhibition rate, the legend’s meaning is consistent with that in (**f**). (**h**) IC50 for α-glucosidase inhibition. (**i**) IC50 for lipase inhibition. Statistical significance among species (*p* < 0.05) is denoted by different letters.

**Figure 5 foods-14-00302-f005:**
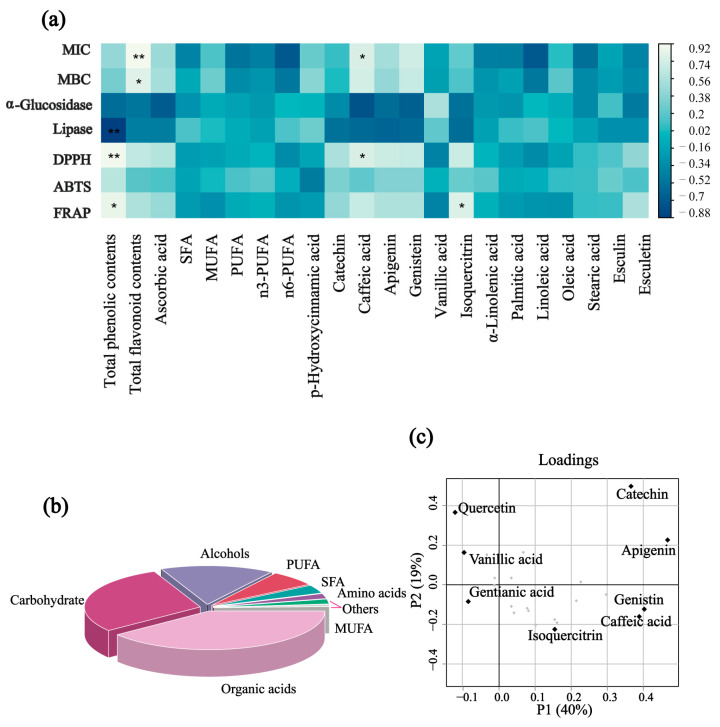
(**a**) Heatmap illustrating the correlation between the bioactivity of the seven *Viola* leaf extracts and their chemical composition. ** denotes a highly significant correlation (*p* < 0.01), while * indicates a significant correlation (*p* < 0.05). (**b**) Pie chart representing the proximate composition of substances in seven *Viola* leaves. (**c**) Loading plot depicting the distribution of phenolic and flavonoid compounds across *Viola* leaves.

**Table 1 foods-14-00302-t001:** The biomass, water content, proximate, and fatty acids composition of the seven *Viola* leaves.

	Va	Vc	Vm	Vp	Vr	Vt	Vv
Biomass (g FW)	7.18 ± 0.47 a	2.09 ± 0.17 e	3.80 ± 0.18 b	3.52 ± 0.04 bc	3.31 ± 0.07 bc	2.62 ± 0.42 de	2.84 ± 0.07 cd
Water content (%)	85.35 ± 4.37 ab	90.96 ± 0.57 ab	84.32 ± 4.64 ab	82.65 ± 2.16 b	86.71 ± 2.68 a	84.43 ± 3.62 ab	85.67 ± 1.18 ab
Proteins (% FW)	5.74 ± 0.17 a	2.18 ± 0.49 c	5.49 ± 0.01 a	2.95 ± 0.18 bc	3.63 ± 0.46 b	1.35 ± 0.27 d	2.37 ± 0.35 c
Soluble sugars (% FW)	7.31 ± 1.14 a	5.14 ± 1.11 c	6.14 ± 0.74 b	4.70 ± 0.19 d	5.25 ± 0.06 c	5.85 ± 0.21 b	5.08 ± 0.42 cd
Total phenolic content (mg GAE/100 g FW)	377.41 ± 105.47 a	191.17 ± 28.10 bc	384.80 ± 8.94 a	144.59 ± 25.53 bc	359.16 ± 93.46 a	131.13 ± 8.51 c	272.26 ± 14.70 ab
Total flavonoid content (mg RE/100 g FW)	26.89 ± 0.27 b	15.64 ± 0.16 de	40.08 ± 0.40 a	13.09 ± 0.13 e	21.89 ± 0.22 c	18.98 ± 0.19 cd	16.60 ± 0.17 de
Carotenoid (mg/100 g FW)	18.62 ± 0.93 c	10.23 ± 0.31 e	4.31 ± 0.05 f	10.12 ± 0.27 e	52.21 ± 0.67 a	13.42 ± 0.51 d	34.31 ± 0.89 b
Chlorophyll (mg/100 g FW)	415.52 ± 4.86 a	250.16 ± 1.92 e	294.47 ± 2.34 d	341.64 ± 1.53 b	101.12 ± 2.07 g	163.66 ± 2.52 f	308.69 ± 1.64 c
Ascorbic acid (mg/100 g FW)	404.12 ± 77.31 a	102.76 ± 23.16 c	259.89 ± 55.56 b	198.94 ± 10.31 b	86.84 ± 8.23 c	58.14 ± 17.25 c	51.89 ± 18.63 c
Fatty acids (μg/g DW)							
Butyric acid	0.03 ± 0.01 e	0.10 ± 0.00 bc	0.12 ± 0.02 a	0.08 ± 0.01 cd	0.06 ± 0.01 d	0.11 ± 0.01 ab	0.03 ± 0.01 e
Hexanoic acid	0.07 ± 0.01 bcd	0.06 ± 0.01 cd	0.08 ± 0.01 bc	0.07 ± 0.00 bcd	0.05 ± 0.01 d	0.08 ± 0.01 b	0.11 ± 0.01 a
Octanoic acid	0.71 ± 0.08 bcd	0.64 ± 0.05 cd	0.96 ± 0.06 b	0.89 ± 0.09 bc	0.60 ± 0.01 d	0.93 ± 0.09 b	3.81 ± 0.27 a
Decanoic acid	0.66 ± 0.07 d	0.55 ± 0.03 d	2.00 ± 0.29 b	1.37 ± 0.06 bc	0.52 ± 0.01 d	2.00 ± 0.16 b	3.05 ± 0.19 a
Undecanoic acid	0.09 ± 0.01 d	0.08 ± 0.00 d	0.17 ± 0.00 bc	0.24 ± 0.04 a	0.09 ± 0.01 d	0.13 ± 0.01 cd	0.20 ± 0.04 ab
Lauric acid	4.52 ± 0.45 c	4.59 ± 0.17 c	18.86 ± 1.09 a	9.43 ± 0.93 b	3.86 ± 0.60 c	8.12 ± 0.74 b	8.79 ± 0.38 b
Tridecanoic acid	0.23 ± 0.02 d	0.25 ± 0.02 d	0.53 ± 0.09 ab	0.61 ± 0.00 a	0.24 ± 0.03 d	0.40 ± 0.01 c	0.51 ± 0.03 b
Myristic acid	29.81 ± 3.24 cd	24.91 ± 1.75 de	86.70 ± 7.55 a	53.55 ± 4.93 b	19.83 ± 0.94 e	30.17 ± 0.59 cd	34.72 ± 1.07 c
Myristoleic acid	1.09 ± 0.16 bc	0.79 ± 0.09 cd	1.79 ± 0.28 a	1.15 ± 0.01 b	0.76 ± 0.10 cd	0.57 ± 0.08 d	1.01 ± 0.16 bc
Pentadecanoic acid	6.75 ± 0.23 f	10.93 ± 0.25 cd	9.69 ± 0.92 ab	14.14 ± 1.63 a	8.55 ± 0.52 e	12.7 ± 0.06 bc	23.59 ± 0.94 a
10Z-Pentadecenoic acid	1.39 ± 0.07 a	0.97 ± 0.10 c	1.31 ± 0.20 ab	1.36 ± 0.12 a	1.15 ± 0.16 abc	1.01 ± 0.05 bc	1.10 ± 0.18 abc
Palmitic acid	1270.85 ± 147.17 b	1187.71 ± 184.28 bc	911.34 ± 41.86 cd	1693.89 ± 193.67 a	962.15 ± 28.85 bcd	778.99 ± 118.71 d	1186.68 ± 163.86 bc
Palmitoleic acid	8.63 ± 0.65 e	22.66 ± 2.08 cd	63.91 ± 9.46 a	25.56 ± 3.30 c	13.26 ± 1.26 de	12.18 ± 1.56 de	46.71 ± 7.21 b
Heptadecanoic acid	17.45 ± 1.71 cd	16.13 ± 0.15 cd	13.82 ± 0.75 d	33.17 ± 0.26 a	15.12 ± 1.69 cd	18.88 ± 1.89 c	25.17 ± 3.09 b
10Z-Heptadecenoic acid	2.58 ± 0.06 b	1.64 ± 0.21 c	0.86 ± 0.14 cd	6.63 ± 0.98 a	1.30 ± 0.01 cd	0.69 ± 0.11 d	0.99 ± 0.02 cd
Stearic acid	138.79 ± 17.28 a	110.70 ± 5.33 bc	83.26 ± 6.86 c	145.07 ± 12.44 a	109.77 ± 6.40 bc	85.65 ± 13.04 bc	111.57 ± 15.93 b
Elaidic acid	0.52 ± 0.05 b	0.37 ± 0.05 bc	0.55 ± 0.06 b	1.21 ± 0.19 a	0.31 ± 0.04 c	0.47 ± 0.03 bc	0.47 ± 0.03 bc
Oleic acid	398.25 ± 43.72 a	358.62 ± 26.78 a	336.81 ± 8.66 a	384.38 ± 62.13 a	198.69 ± 11.93 b	240.64 ± 38.19 b	350.21 ± 5.88 a
Linolelaidic acid	0.20 ± 0.02 d	0.21 ± 0.00 d	0.31 ± 0.00 b	0.27 ± 0.01 c	0.34 ± 0.04 b	0.14 ± 0.01 e	0.40 ± 0.01 a
Linoleic acid	925.39 ± 34.21 c	929.49 ± 27.38 c	628.07 ± 52.30 d	1336.68 ± 168.99 a	891.7 ± 85.67 c	656.71 ± 14.74 d	1153.11 ± 42.29 b
γ-Linoleic acid	0.79 ± 0.05 a	0.68 ± 0.11 b	0.57 ± 0.01 c	0.85 ± 0.04 a	0.39 ± 0.01 d	0.31 ± 0.02 d	0.54 ± 0.06 c
α-Linolenic acid	2338.52 ± 368.05 ab	1843.87 ± 227.83 c	1141.24 ± 153.03 d	2580.16 ± 283.22 a	1291.50 ± 146.93 d	571.02 ± 65.01 e	2067.12 ± 84.05 bc
Arachidic acid	8.12 ± 0.11 d	7.71 ± 0.57 d	23.13 ± 0.32 a	21.24 ± 2.67 a	16.52 ± 0.10 b	12.90 ± 0.68 c	17.02 ± 1.67 b
11Z-Eicosenoic acid	13.65 ± 0.55 b	12.77 ± 0.12 b	11.36 ± 0.10 b	26.11 ± 4.21 a	14.18 ± 0.01 b	12.51 ± 1.12 b	14.02 ± 1.57 b
11Z,14Z-Eicosadienoic acid	8.57 ± 0.90 c	10.57 ± 0.43 b	6.15 ± 0.61 d	14.33 ± 0.50 a	10.79 ± 1.61 b	6.04 ± 0.01 d	12.61 ± 0.55 a
8Z,11Z,14Z-Eicosatrienoic acid	0.06 ± 0.00 e	0.11 ± 0.01 de	1.15 ± 0.07 b	0.30 ± 0.02 d	0.78 ± 0.07 c	0.28 ± 0.01 d	1.55 ± 0.22 a
11Z,14Z,17Z-Eicosatrienoic acid	9.43 ± 0.36 b	6.83 ± 1.09 c	4.39 ± 0.31 de	11.99 ± 1.45 a	3.81 ± 0.12 e	1.69 ± 0.16 f	5.89 ± 0.62 cd
Arachidonic acid	0.35 ± 0.01 c	0.52 ± 0.03 c	2.28 ± 0.14 a	0.85 ± 0.15 b	0.86 ± 0.10 b	0.88 ± 0.02 b	2.27 ± 0.09 a
5Z,8Z,11Z,14Z,17Z-Eicosapentaenoic acid	7.51 ± 0.04 c	8.42 ± 0.50 c	14.26 ± 2.01 a	13.21 ± 0.98 a	12.36 ± 1.16 ab	7.99 ± 0.19 c	10.88 ± 0.12 b
Heneicosanoic acid	1.58 ± 0.06 d	1.72 ± 0.00 cd	3.51 ± 0.20 a	2.52 ± 0.29 b	1.92 ± 0.20 cd	2.02 ± 0.12 c	3.34 ± 0.21 a
Behenic acid	0.78 ± 0.01 c	1.18 ± 0.10 bc	1.97 ± 0.32 a	1.04 ± 0.14 bc	0.78 ± 0.07 c	1.41 ± 0.22 b	2.07 ± 0.29 a
Erucic acid	1.54 ± 0.22 b	1.94 ± 0.31 a	2.16 ± 0.02 a	2.18 ± 0.01 a	0.85 ± 0.08 c	1.01 ± 0.10 c	2.14 ± 0.12 a
13Z,16Z-Docosadienoic acid	7.39 ± 0.67 b	4.37 ± 0.36 c	6.84 ± 0.84 b	6.95 ± 0.67 b	4.75 ± 0.02 c	8.64 ± 0.90 b	10.74 ± 1.40 a
Adrenic acid	nd	nd	nd	nd	nd	nd	nd
4Z,7Z,10Z,13Z,16Z-Docosapentaenoic acid	0.50 ± 0.08 b	0.74 ± 0.08 a	0.35 ± 0.05 c	0.61 ± 0.05 b	0.60 ± 0.11 ab	0.68 ± 0.02 a	0.33 ± 0.01 c
4Z,7Z,10Z,13Z,16Z,19Z-Docosahexaenoic acid	2.23 ± 0.28 c	2.23 ± 0.09 c	3.98 ± 0.05 b	1.84 ± 0.04 c	1.65 ± 0.19 c	4.39 ± 0.19 b	5.23 ± 0.85 a
Tricosanoic acid	15.40 ± 0.10 b	16.28 ± 0.66 b	14.20 ± 1.46 b	22.60 ± 1.01 a	14.84 ± 2.28 b	12.62 ± 2.04 b	19.81 ± 2.16 a
Lignoceric acid	56.52 ± 4.07 b	61.14 ± 6.94 b	40.95 ± 1.14 c	92.17 ± 8.09 a	59.59 ± 6.46 b	39.80 ± 1.71 c	56.38 ± 4.37 b
Nervonic acid	3.07 ± 0.45 a	2.45 ± 0.10 b	0.95 ± 0.03 cd	2.90 ± 0.31 ab	1.29 ± 0.14 c	0.89 ± 0.13 cd	0.69 ± 0.07 d
SFA	1552.34 ± 7.45 b	1444.68 ± 221.93 bc	1211.27 ± 65.53 cd	2092.09 ± 66.78 a	1214.50 ± 168.20 cd	1006.91 ± 66.66 d	1496.86 ± 226.17 bc
MUFA	430.72 ± 65.16 a	402.20 ± 26.16 a	419.71 ± 17.56 a	451.49 ± 15.61 a	231.78 ± 4.19 b	269.96 ± 0.46 b	417.34 ± 24.94 a
PUFA	3303.09 ± 114.30 b	2809.92 ± 168.05 c	1810.83 ± 95.86 e	3970.50 ± 118.30 a	2220.49 ± 64.52 d	1259.9 ± 169.18 f	3271.66 ± 2.82 b
ω3-PUFA	2359.84 ± 92.91 ab	1863.22 ± 61.97 c	1165.12 ± 80.90 d	2609.65 ± 259.91 a	1310.28 ± 66.29 d	586.21 ± 39.08 e	2090.11 ± 209.45 bc
ω6-PUFA	943.25 ± 125.11 c	946.70 ± 36.52 c	645.71 ± 83.74 d	1360.85 ± 46.55 a	910.20 ± 17.40 c	673.69 ± 72.36 d	1181.54 ± 73.04 b

Data are expressed as mean ± SD on fresh weight basis (*n* = 3). Different letters indicate significant differences (*p* < 0.05) among treatment means as determined by Duncan’s test. nd—not detected, indicating that the substance was below the detection limit of the analytical method used. DW—dry weight; FW—fresh weight. Va—*Viola acuminata*; Vc—*Viola collina*; Vm—*Viola mirabilis*; Vp—*Viola philippica*.*;* Vr—*Viola prionantha;* Vt—*Viola tokubuchiana* var*. takedana;* Vv—*Viola variegata*.

**Table 2 foods-14-00302-t002:** The phenolic, flavonoid, and coumarin compounds of the seven *Viola* leaves.

No.	Compound Name (μg/kg FW)	Va	Vc	Vm	Vp	Vr	Vt	Vv
1	p-Hydroxycinnamic acid	365.47 ± 51.87 a	175.93 ± 48.13 b	333.67 ± 9.73 a	394.67 ± 16.73 a	131.53 ± 41.93 b	210.67 ± 64.47 b	404.20 ± 14.40 a
2	Benzoic acid	349.27 ± 95.60 b	336.13 ± 50.07 b	417.07 ± 97.80 b	390.8 ± 44.8 b	297.47 ± 43.00 b	171.00 ± 22.93 b	771.47 ± 251.20 a
3	Gentianic acid	49.20 ± 4.80 c	84.73 ± 27.93 bc	77.20 ± 10.27 c	143.53 ± 29.27 ab	87.60 ± 45.80 bc	167.80 ± 4.60 a	58.87 ± 23.60 c
4	Eugenic acid	22.07 ± 7.87 c	93.93 ± 15.40 a	56.67 ± 5.13 b	46.80 ± 6.93 b	9.53 ± 1.20 c	8.00 ± 1.13 c	41.67 ± 5.67 b
5	Cinnamic acid	126.53 ± 21.67 bc	69.67 ± 16.73 bc	316.73 ± 96.53 a	95.20 ± 24.13 bc	155.73 ± 25.80 b	33.53 ± 11.93 c	130 ± 49.07 bc
6	Caffeic acid	525.73 ± 102.47 b	67.80 ± 14.20 dc	724.60 ± 53.40 a	177.60 ± 20.60 d	332.20 ± 43.80 c	94.13 ± 11.20 dc	55.67 ± 4.80 e
7	Procatechin	19.93 ± 5.07 a	0.93 ± 0.20 d	11.07 ± 1.67 bc	3.53 ± 1.27 cd	13.40 ± 4.73 ab	2.87 ± 1.13 cd	21.40 ± 5.47 a
8	Vanillic acid	99.07 ± 25.40 bc	112.13 ± 20.67 c	192.80 ± 18.73 b	126.27 ± 36.60 c	40.27 ± 19.80 d	125.93 ± 23.00 c	689.60 ± 33.73 a
9	Rosmarinic acid	32.80 ± 8.27 ab	18.53 ± 4.33 b	17.93 ± 3.80 b	34.60 ± 8.00 ab	49.80 ± 20.73 a	49.60 ± 7.40 a	18.87 ± 7.87 b
10	Mustelic acid	4.07 ± 1.47 b	6.53 ± 3.07 b	84.00 ± 10.53 a	0.73 ± 0.07 b	11.47 ± 4.73 b	2.87 ± 0.53 b	6.60 ± 5.87 b
11	Genistein	316.73 ± 118.13 ab	193.87 ± 51.53 bc	435.40 ± 29.87 a	46.27 ± 29.27 d	90.40 ± 49.20 cd	24.33 ± 11.47 d	5.27 ± 1.87 d
12	Apigenin	1016.53 ± 126.87 a	164.07 ± 73.4 c	543.33 ± 36.93 b	36.8 ± 23.00 cd	80.87 ± 38.73 cd	11.60 ± 7.47 d	13.53 ± 7.80 d
13	Naringin	134.40 ± 35.07 b	19.53 ± 5.33 c	213.20 ± 18.93 a	6.67 ± 1.40 c	39.80 ± 14.60 c	7.67 ± 1.00 c	19.20 ± 2.40 c
14	Quercetin	402.00 ± 30.4 bc	514.87 ± 27.40 a	132.87 ± 10.00 d	418.60 ± 11.20 bc	341.27 ± 80.60 c	196.13 ± 8.47 d	486.27 ± 38.73 ab
15	Catechin	1238.47 ± 246.33 a	55.67 ± 11.73 b	53.07 ± 11.67 b	62.40 ± 5.73 b	79.47 ± 10.80 b	54.67 ± 8.00 b	66.13 ± 19.80 b
16	Kaempferol	36.47 ± 9.33 bc	6.20 ± 1.53 c	56.73 ± 16.47 b	18.47 ± 15.53 bc	180.27 ± 33.27 a	11.93 ± 6.33 c	21.27 ± 2.47 bc
17	Glycyrrhizin	0.73 ± 0.33 c	7.07 ± 3.33 bc	4.87 ± 1.40 abc	2.07 ± 1.33 bc	4.67 ± 3.73 abc	1.67 ± 1.00 bc	9.13 ± 2.20 a
18	Galangin	184.20 ± 9.47 a	60.47 ± 18.67 b	187.53 ± 15.80 a	15.13 ± 8.20 c	21.20 ± 9.60 c	5.60 ± 2.33 c	2.67 ± 0.33 c
19	Rutin	15.20 ± 3.33 bc	1.60 ± 0.67 c	73.47 ± 8.47 a	3.07 ± 1.00 c	25.87 ± 12.60 b	14.67 ± 0.80 bc	18.27 ± 2.60 b
20	Genistin	562.67 ± 201.33 a	23.07 ± 10.00 b	665.80 ± 37.07 a	25.27 ± 15.87 b	124.47 ± 64.07 b	145.87 ± 123.67 b	12.53 ± 2.07 b
21	Echinacoside	183.27 ± 21.53 bc	65.73 ± 8.20 d	421.40 ± 42.47 a	89.73 ± 6.20 cd	40.93 ± 4.47 d	200.47 ± 105.47 b	197.27 ± 13.00 bc
22	Isoquercitrin	338.13 ± 103.40 c	56.47 ± 12.07 d	509.87 ± 46.07 b	169.67 ± 19.53 d	837.80 ± 14.67 a	125.47 ± 32.47 d	320.00 ± 42.67 c
23	Esculin	60,446.09 ± 9115.45	nd	nd	8282.43 ± 801.62	nd	nd	57,788.55 ± 1873.16
24	Esculetin	nd	nd	nd	4062.15 ± 44.25	10,502.51 ± 1399.32	nd	nd

Data represent the mean values for each sample ± standard deviation (*n* = 3). Different letters in a, b, c and d indicate significant differences between different ferns by Duncan’s test (*p* < 0.05). Detection limit, 0.0002 mg/g. nd: not detected.

**Table 3 foods-14-00302-t003:** The antimicrobial activity of the seven Viola leaf extracts against *Staphylococcus aureus*, Escherichia coli, *Pseudomonas aeruginosa*, and *Candida albicans*.

Species	Antibacterial Activity									Antifungal Activity		
	Gram+			Gram−								
	*S. aureus*			*E. coli*			*P. aeruginosa*			*C. albicans*		
	Inhibition Zone Diameter	MIC	MBC	Inhibition Zone Diameter	MIC	MBC	Inhibition Zone Diameter	MIC	MBC	Inhibition Zone Diameter	MIC	MBC
	250 mg/mL	(mg/mL)	(mg/mL)	250 mg/mL	(mg/mL)	(mg/mL)	250 mg/mL	(mg/mL)	(mg/mL)	250 mg/mL	(mg/mL)	(mg/mL)
Va	7.60 ± 0.30 b	125	125	7.61 ± 0.27 cd	125	125	7.91 ± 0.50 a	125	125	7.94 ± 0.29 ab	62.5	125
Vc	8.61 ± 0.47 a	62.5	62.5	7.76 ± 1.15 cd	125	125	8.19 ± 0.66 a	125	125	7.39 ± 0.10 bc	62.5	62.5
Vm	6.89 ± 0.19 c	>250	>250	8.80 ± 0.07 bc	>250	>250	7.78 ± 0.05 a	>250	>250	8.55 ± 0.14 a	>250	>250
Vp	9.01 ± 0.11 a	62.5	125	11.09 ± 1.10 a	125	250	7.80 ± 0.97 a	125	250	7.26 ± 1.16 bc	62.5	62.5
Vr	8.57 ± 0.07 a	62.5	62.5	7.53 ± 0.39 d	>250	>250	8.23 ± 0.16 a	>250	>250	7.29 ± 0.43 bc	125	125
Vt	7.04 ± 0.30 c	125	125	8.37 ± 0.38 cd	125	>250	8.55 ± 0.08 a	>250	>250	6.88 ± 0.11 c	125	125
Vv	8.73 ± 0.63 a	62.5	62.5	9.92 ± 0.50 b	62.5	62.5	8.22 ± 0.57 a	62.5	125	8.44 ± 0.69 a	62.5	62.5
Sterile water	6.00 ± 0.00	nd	nd	6.00 ± 0.00	nd	nd	6.00 ± 0.00	nd	nd	6.00 ± 0.00	nd	nd
Ampicillin	nd	<10	<10	nd	10	10	nd	20	>20	nd	nd	nd
Kanamycin	10.79 ± 0.02	nd	nd	9.06 ± 0.41	nd	nd	9.97 ± 0.41	nd	nd	6.79 ± 0.41	nd	nd

Values are expressed as mean ± SD with three replications (*n* = 3) for inhibition zone diameter data. Different letters in the same column indicate significant differences (*p* < 0.05). Inhibition zone diameter represents the clear area around a sample where bacterial growth is inhibited, measured in millimeters. nd: not detected. MIC—minimal inhibitory concentration; MBC—minimal bactericidal concentration. The concentration of ampicillin was 0.25 × 10^−3^ g/mL, and the concentration of kanamycin was 0.25 × 10^−3^ g/mL, both dissolved in ethanol. *S. aureus*—*Staphylococcus aureus*; *E. coli*—*Escherichia coli*; *P. aeruginosa*—*Pseudomonas aeruginosa*; *C. albican*—*Candida albicans*.

## Data Availability

The original contributions presented in the study are included in the article/Appendix A, further inquiries can be directed to the corresponding authors.

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
