# Peer review of "Proximate Composition and In Vitro Bioactive Properties of Leaf Extracts from Seven Viola Species"

_foods, 2025, doi:10.3390/foods14020302_

Round 1
Reviewer 1 Report
Comments and Suggestions for Authors
The present contribution by Zeng et al. considered the proximate composition and biological activities of the ethanol extract of seven wild Viola species. The work is detailed and provide valuable information that could constitute a theoretical basis for further investigation into these wild Viola species as well as their integration into the food system as potential nutraceuticals or functional food ingredients. Authors are also worthy of commendation for a refreshing presentation. Although this work could constitute a valuable contribution, there is still room for improvement. Authors are encourage to clearly describe the materials and methods used in the investigation and further substantiate the results by providing a more robust explanation for the biological effects. Also, comparison with recent and similar published work would be valuable. For minor comments, please consider the issues stated below.
1) Title: It should be indicated in the title that the work pertains to ‘seven Viola species’. Authors should also state in the title that the work is about the leaves extract. No benefit can be derived from being vague and ambiguous with the Title.
2) The numerical value of the crude protein content should be included in the Abstract.
3) Authors should also mention the extract concentration that demonstrated the respective α-glucosidase and lipase inhibitory activities.
4) Lines 15-16: ‘strongly associated’ does not convey any particular meaning. Authors should be more specific in describing the relationship.
4) Lines 17-18 “These results indicate that Viola offers notable health benefits, presenting a valuable addition to enhancing modern dietary patterns and overall health.”
Authors should be more cautious and responsible with the use of language in scientific communication. There is NO indication in this work that the leaves extract “offers notable health benefits…” This aspect of the Abstract should be corrected.
6) Lines 38-42: Authors should be specific when describing the content and properties of botanical materials. What particular plant, plant part, extract, solvent used in the extraction, etc.? These details should be included.
7) Section 2.1, Line 72. Authors should also include the name of the expert who identified/authenticated the Viola samples, the voucher numbers and name of the Herbarium where reference specimens were deposited.
8) Line 101. Physiological?
9) Most of the methods were not described in any meaningful detail. Thus, it would not be possible for readers or other researchers to reproduce. Authors are required to provide complete step-by-step description of the sample preparation, methods used as well as equipment/apparatus.
10) Line 193 ‘0.19 g to 1.04 g…per what?
11) Lines 199-201: ‘Typical protein levels in most vegetables hover around 3.0 g/100 g FW, suggesting Viola leaves are a robust plant-based protein source with potential to meet human dietary protein needs, supporting prospects for their development and utilization.’
The above sentences are technically inaccurate and should be revise. A food material with 3% w/w protein is not your typical robust protein source. Perhaps the authors should seek clarification from the UNO/FAO/WHO expert committee report for protein dietary requirement.
12) Lines 111 and 206-211: Authors should clarify whether the values of TPC and TPC were presented in mg/g dry weight of the extract since it was not mentioned that the extract was dried prior to the analysis.
13) Line 222. Physiological? There is nothing physiological in the table. Authors should correct it.
14) Figure 3 is of very poor quality and the details are not visible. It should be replaced.
15) Section 3.2.3. Enzyme activity inhibition capacity.
Authors should further expatiate on the rationale underlying the differences in enzyme inhibitory activity with emphasis on the potential relationship between the phyto-constituents and biological activity.
16) Authors should indicate weakness and limitations associated with this work and how that could impact translating the findings into real-world application.
17) Authors should meticulously edit this manuscript for English language errors.
Thanks and good luck!
Comments on the Quality of English LanguageThis manuscript needs English language editing.
Author Response
The present contribution by Zeng et al. considered the proximate composition and biological activities of the ethanol extract of seven wild Viola species. The work is detailed and provide valuable information that could constitute a theoretical basis for further investigation into these wild Viola species as well as their integration into the food system as potential nutraceuticals or functional food ingredients. Authors are also worthy of commendation for a refreshing presentation. Although this work could constitute a valuable contribution, there is still room for improvement. Authors are encourage to clearly describe the materials and methods used in the investigation and further substantiate the results by providing a more robust explanation for the biological effects. Also, comparison with recent and similar published work would be valuable. For minor comments, please consider the issues stated below.
- Title: It should be indicated in the title that the work pertains to ‘seven Viola species’. Authors should also state in the title that the work is about the leaves extract. No benefit can be derived from being vague and ambiguous with the Title.
A: Thank you for your valuable suggestions.
We changed the title as your suggestions to “Proximate composition and in vitro bioactive properties of leaf extracts from seven Viola species”
- The numerical value of the crude protein content should be included in the Abstract.
A: Thanks for your comments.
We are very sorry for our oversight in using incorrect calculation units for the protein content results. We strictly calculated the crude protein content based on the original data again, the results showed that the crude protein content vary from 1.35 % to 5.74 % FW. Considering the United Nations report on dietary protein requirements, they cannot be considered a high-protein edible material. Therefore, we deleted the sentence in the abstract.
- Authors should also mention the extract concentration that demonstrated the respective α-glucosidase and lipase inhibitory activities.
A: Thank you for your suggestion.
We added the extraction concentration in the respective α-glucosidase and lipase inhibitory activities in the abstract part.
- Lines 15-16: ‘strongly associated’ does not convey any particular meaning. Authors should be more specific in describing the relationship.
A: Thanks for your comments. We changed it.
- Lines 17-18 “These results indicate that Viola offers notable health benefits, presenting a valuable addition to enhancing modern dietary patterns and overall health.” Authors should be more cautious and responsible with the use of language in scientific communication. There is NO indication in this work that the leaves extract “offers notable health benefits…” This aspect of the Abstract should be corrected.
A: Thanks for your valuable comments.
In this paper, we investigated the antioxidant activity, antimicrobial activity and enzyme inhibition capacity of Viola leaves extraction. All of the results confirmed the extraction have the potential benefits for the health. Therefore, we changed the sentence to “these results indicate that Viola offers potential notable health benefits, presenting a valuable addition to enhancing modern dietary patterns and overall health.”
- Lines 38-42: Authors should be specific when describing the content and properties of botanical materials. What particular plant, plant part, extract, solvent used in the extraction, etc.? These details should be included.
A: Thank you for your valuable comments.
We added details according to references in the introduction part.
- Section 2.1, Line 72. Authors should also include the name of the expert who identified/authenticated the Viola samples, the voucher numbers and name of the Herbarium where reference specimens were deposited.
A: Thank you for your comments. We have added the name of expert who identified the Viola samples.
- Line 101. Physiological?
A: Thanks for your comments.
By carefully check the manuscript, there were not contents about Physiological in section 2.3. Therefore, we deleted this character.
- Most of the methods were not described in any meaningful detail. Thus, it would not be possible for readers or other researchers to reproduce. Authors are required to provide complete step-by-step description of the sample preparation, methods used as well as equipment/apparatus.
A: Thank you for your comments.
We have thoroughly revised Lines 96-189 to provide a detailed, step-by-step description of the sample preparation, methods used, and the equipment/apparatus involved according to your suggestions.
- Line 193 ‘0.19 g to 1.04 g…per what?
A: Thank you for your comments.
We dried the fresh Viola by keeping them in the oven at 60 ℃. After that, we weighed the weight of each plant. Thus, the unit in here should be 0.19 g to 1.04 g per whole Viola plant. We changed it accordingly.
- Lines 199-201: ‘Typical protein levels in most vegetables hover around 3.0 g/100 g FW, suggesting Viola leaves are a robust plant-based protein source with potential to meet human dietary protein needs, supporting prospects for their development and utilization.’ The above sentences are technically inaccurate and should be revise. A food material with 3% w/w protein is not your typical robust protein source. Perhaps the authors should seek clarification from the UNO/FAO/WHO expert committee report for protein dietary requirement.
A: Thank you for your valuable comments.
As discussed in the Answer 2, we check the clarification from the United Nations and the World Health Organization on dietary protein requirements, the Viola leaves of cannot meet human dietary protein needs. Hence, we deleted the sentence and newly added the discussion of soluble sugar.
- Lines 111 and 206-211: Authors should clarify whether the values of TPC and TPC were presented in mg/g dry weight of the extract since it was not mentioned that the extract was dried prior to the analysis.
A: Thank you for your valuable suggestions.
The unit of TPC should be mg GAE/100 g FW. We check the manuscript thoroughly and changed it accordingly.
- Line 222. Physiological? There is nothing physiological in the table. Authors should correct it.
A: Thanks for your comments.
We corrected it and changed the name of title to “Biomass, water content, proximate and fatty acids composition of seven Viola species.”
- Figure 3 is of very poor quality and the details are not visible. It should be replaced.
A: Thank you for your comments.
We have replaced Figure 3 with high quality files.
- Section 3.2.3. Enzyme activity inhibition capacity. Authors should further expatiate on the rationale underlying the differences in enzyme inhibitory activity with emphasis on the potential relationship between the phyto-constituents and biological activity.
A: Thank you for your valuable comments.
Previous studies suggested that phytoconstituents (phenolic compounds) could inhibit α-glucosidase and pancreatic lipase through competitive binding at the enzyme's active site or by altering its conformation through hydrogen bonding and π-π interactions. Moreover, the antioxidant properties of these phytoconstituents may further stabilize the enzyme-substrate complex, enhancing inhibitory efficiency. We added this discussion in the section 3.2.3.
- Authors should indicate weakness and limitations associated with this work and how that could impact translating the findings into real-world application.
A: Thanks for your valuable suggestions.
Although we have demonstrated the wild edible plant Viola is a rich source of essential nutrients, including protein, organic acids, phenolic and flavonoid compounds, as well as ω-3 fatty acids like EPA and DHA, which are beneficial for health. Future studies are encouraged to isolate and characterize individual bioactive compounds from Viola species to gain a deeper understanding of their specific mechanisms of action and to explore their potential applications in functional foods, nutraceuticals, and pharmaceuticals. We added this discussion in the conclusion part.
- Authors should meticulously edit this manuscript for English language errors.
A: Thank you very much for your suggestion.
We sincerely apologize for the limitations in our English writing skills. We have meticulously reviewed and extensively revised the manuscript to address English language issues. We have made significant efforts to improve clarity, grammar, and overall readability.
Reviewer 2 Report
Comments and Suggestions for Authors
The title should be more explicit and be consistent with the results obtained.
The abstract must consist of an introduction, objective, methodology, results and conclusions.
In general, the methodology must be more detailed so that it can be reproduced.
The results must be discussed in detail.
The correlations of the results obtained need to be discussed in greater depth, since it seems that it is a report of analysis and biological activities found.
It is important, with the results obtained, to be able to elucidate the mechanisms that contribute to antioxidant and antimicrobial activity.
In the conclusion, it is important to conclude and not summarize results. Furthermore, it is relevant to provide perspectives for future work.
Author Response
- The title should be more explicit and be consistent with the results obtained.
A: Thank you for your comments.
We changed the title to “Proximate composition and in vitro bioactive properties of leaf extracts from seven Viola species”.
- The abstract must consist of an introduction, objective, methodology, results and conclusions.
A: Thanks for your comments.
As your suggestions, we have reorganized the abstract.
- In general, the methodology must be more detailed so that it can be reproduced.
A: Thank you for your valuable comments.
As your suggestions, we have thoroughly revised the methodology part, and added the detailed, step-by-step description of the sample preparation, methods, and the equipment/apparatus involved, to clarify the procedures and make them easier to reproduce.
- The results must be discussed in detail.
A: Thanks for your suggestions.
We appreciate your suggestion to provide a more detailed discussion of the results. To facilitate a more in-depth discussion, we have expanded the discussion section to include a more comprehensive analysis and interpretation of the findings, incorporating relevant literature to enhance the depth and clarity. We marked the newly added content with red in the revised manuscript.
- The correlations of the results obtained need to be discussed in greater depth, since it seems that it is a report of analysis and biological activities found.
A: Thanks for your suggestions.
We added the discussion accordingly in the Result and discussion part. All of the changes were marked with red in the revised manuscript.
- It is important, with the results obtained, to be able to elucidate the mechanisms that contribute to antioxidant and antimicrobial activity.
A: Thanks for your valuable comments.
According to our results, the leaves extraction of Viola is rich in phenolic compounds, and contains various flavonoids, fatty acids, and essential minerals. Among them, phenolic compounds and flavonoids might contribute to antioxidant and antimicrobial activity, due to the abundant phenolic hydroxyl groups in their structure which could scavenge radicals and destroy the cell membrane (Chemistry & Biodiversity, 2019, 16(12): e1900426.).
- In the conclusion, it is important to conclude and not summarize results. Furthermore, it is relevant to provide perspectives for future work.
A: Thank you for your comments.
We have revised the conclusion section to focus on providing a concise conclusion rather than summarizing the results. Additionally, we have gave our perspectives for future research to address potential directions and applications.
Reviewer 3 Report
Comments and Suggestions for Authors
- The title must be revised to reflect the research focus, such as mentioning "Viola leaves." or viola extracts.. etc. also, apply this issue throughout teh mansucript, don not write viola without any specification.
- Lines 12–13: The writing is confusing. Additionally, define abbreviations the first time they are mentioned.
- The abstract is poorly written. Include numerical data and significant values to improve clarity and impact.
- "Essential minerals" is too broad; specify which minerals are being discussed.
- The study's main insights should be emphasized with more specific wording.
- Specify the Viola genus or species studied in the title and throughout the manuscript.
- In the title and abstract, clearly mention the "7 Viola species" studied.
- Line 97: Provide more details about the extraction process, centrifuge parameters, model of the instrument, etc.
- Why was ethanol used for extraction? Why not methanol or aqueous methanol? Clarify your reasoning.
- Line 111: Add a reference for total phenol content.
- Provide more details on untargeted primary metabolite profiling of plant samples using GC-MS, including the column, instrumentation used. Add their country, and models..etc.
- Explain how quantification was performed.
- The section on characterization of phenolic and flavonoid compounds by LC-MS needs to be extended with additional details.
- Revise the titles and captions of figures and tables to specify whether they pertain to Viola leaves, Viola extracts, or another context.
- Add high-quality figures to improve the visual representation of your findings.
- The methods section must be improved to ensure the study can be replicated.
- Table 3: The title, "Antimicrobial activity of Viola," is not accurate. Specify the bacterial names tested.
- Define "Inhibition zone[40]" clearly. Add zone of inhibition data (e.g., images of Petri dishes) as supplementary material.
- Extend the discussion section with recent and related reports, such as:
- Underutilized edible Himalayan herb, Viola canescens Wall.; chemical composition, antioxidant and antimicrobial activity against respiratory tract pathogens
- Main Concern: The cytotoxicity of the Viola extracts was not evaluated, which raises safety concerns. It is recommended to conduct cytotoxicity tests using standard assays such as the MTT or LDH assay.
- Include a detailed methodology for cytotoxicity testing, reporting IC50 values or cell viability percentages, and discussing the safety implications of the extracts.
- Provide data demonstrating whether the extracts are safe for potential applications in medicine, food, or cosmetics.
Expand the antimicrobial testing to include other food-borne pathogens could be interesting.
Author Response
- The title must be revised to reflect the research focus, such as mentioning "Viola leaves." or Viola extracts.. etc. also, apply this issue throughout teh mansucript, don not write Viola without any specification.
A: Thank you for your valuable suggestions.
We changed the title to “Proximate composition and in vitro bioactive properties of leaf extracts from seven Viola species”. In addition, we check the manuscript carefully, and made the description more specification. All the changes were marked with red in the revised manuscript.
- Lines 12–13: The writing is confusing. Additionally, define abbreviations the first time they are mentioned.
A: Thanks for your comments.
We added the definition of abbreviations and reorganized the abstract.
- The abstract is poorly written. Include numerical data and significant values to improve clarity and impact.
A: Thank you for your comments.
We reorganized the abstract by incorporating key quantitative findings and emphasizing the most significant results of the study to enhance the overall quality and precision of the abstract.
- "Essential minerals" is too broad; specify which minerals are being discussed.
A: Thanks for your valuable comments.
As your suggestions, we changed the “essential minerals” to “essential minerals such as potassium, calcium, and magnesium” to make it more specific.
- The study's main insights should be emphasized with more specific wording.
A: Thanks for your comments.
As your suggestion, we have rewritten the abstract to emphasize the study's main insights with clearer and more specific wording, incorporating key quantitative findings to enhance its clarity and impact.
- Specify the Viola genus or species studied in the title and throughout the manuscript.
A: Thank you for your comments.
As discussed in Answer 1, to make the description more specification, we changed the title. Also, we have specified the seven Viola species throughout the manuscript, including main text, figures, and tables to ensure clarity.
- In the title and abstract, clearly mention the "7 Viola species" studied.
A: Thank you for your suggestion.
We added “7 Viola species” in the title and abstract in the revised manuscript.
- Line 97: Provide more details about the extraction process, centrifuge parameters, model of the instrument, etc.
A: Thanks for your comments.
We added more details in the materials and method part. All the changes were marked with red in the revised manuscript.
- Why was ethanol used for extraction? Why not methanol or aqueous methanol? Clarify your reasoning.
A: Thank you for your valuable comments.
Firstly, aqueous ethanol is an excellent solvent for the component’s extraction, and commonly used for extracting plant metabolites due to its effectiveness in isolating polar compounds such as polyphenols and flavonoids. Moreover, the combination of ethanol and water has strong permeability, enabling better dissolution and extraction of target compounds with antioxidant activity. Additionally, compare to methanol or aqueous methanol, ethanol has lower toxicity and low operation requirements. Therefore, aqueous ethanol was chosen in our work.
- Add a reference for total phenol content.
A: Thanks for your comments.
We added a reference for phenol content in Line 127 in the revised manuscript.
- Provide more details on untargeted primary metabolite profiling of plant samples using GC-MS, including the column, instrumentation used. Add their country, and models..etc.
A: Thank you for your valuable suggestions.
As your suggestions, we added the detailed descriptions of the GC-MS non-targeted analysis in the Methods section, including information on the instrument signal, manufacturer, column specifications, and other relevant parameters. These details have been added in Line159-163 in the revised manuscript.
- Explain how quantification was performed.
A: Thank you for your comments.
The quantification method used in our study is based on the previous research conducted by our group. To provide more comprehensive understanding of our experimental method, we have added an additional reference after careful consideration (J. Plant Growth Regul. 41, 2093–2107 (2022)). For the quantification process, calibration curves were established by analyzing the corresponding compounds at different concentrations in parallel with each experimental run. The concentration of the compound was plotted against the peak area to generate the standard curve. The actual concentration of the compounds in the samples was then calculated by substituting the peak area obtained from the sample analysis into the calibration curve. That is the process by which quantification was performed.
- The section on characterization of phenolic and flavonoid compounds by LC-MS needs to be extended with additional details.
A: Thank you for your valuable comment.
We have provided additional detailed descriptions of the experimental methods in the section on the characterization of phenolic and flavonoid compounds by LC-MS.
- Revise the titles and captions of figures and tables to specify whether they pertain to Viola leaves, Viola extracts, or another context.
A: Thanks for your comments.
We carefully revised the titles and captions of figures and tables. To make them more specification, we changed the titles and captions accordingly. All the changes were marked with red in the revised manuscript.
- Add high-quality figures to improve the visual representation of your findings.
A: Thank you for your comments.
To improve the visual representation, we replaced all of the figures by re-output high quality files.
- The methods section must be improved to ensure the study can be replicated.
A: Thanks for your comments.
We carefully revised the method section, and added the additional details and improvements according to your suggestions, to ensure that our experimental methods are clear and reproducible for other researchers.
- Table 3: The title, "Antimicrobial activity of Viola," is not accurate. Specify the bacterial names tested.
A: Thank you for your valuable suggestion. We changed it accordingly.
- Define "Inhibition zone[40]" clearly. Add zone of inhibition data (e.g., images of Petri dishes) as supplementary material.
A: Thanks for your comments.
We added the definition of inhibition zone in the note of Table 3. Additionally, we added the inhibition data as Figure S3 in supplementary materials.
- Extend the discussion section with recent and related reports, such as: Underutilized edible Himalayan herb, Viola canescens Wall.; chemical composition, antioxidant and antimicrobial activity against respiratory tract pathogens
A: Thanks for your suggestions.
We added this discussion and cited the related reports in the results and discussion part. Also, we highlighted the significance of Viola species in addressing various health-related applications. All of the changes were marked with red in the revised manuscript.
- The cytotoxicity of the Viola extracts was not evaluated, which raises safety concerns. It is recommended to conduct cytotoxicity tests using standard assays such as the MTT or LDH assay. Include a detailed methodology for cytotoxicity testing, reporting IC50 values or cell viability percentages, and discussing the safety implications of the extracts. Provide data demonstrating whether the extracts are safe for potential applications in medicine, food, or cosmetics. Expand the antimicrobial testing to include other food-borne pathogens could be interesting.
A: Thank you for your comments. The cytotoxicity assessment would significantly enhance our discussion in the safety implications of the extracts. However, due the fresh Viola materials were collected in the spring, we are unable to conduct these experiments for the raw materials limitation. We truly appreciate your insightful suggestions.
Round 2
Reviewer 2 Report
Comments and Suggestions for Authors
Accept in present form
Author Response
A: Thank you very much for your positive decision to accept our manuscript in its current form. We greatly appreciate the time and effort you invested in reviewing our work.
Reviewer 3 Report
Comments and Suggestions for Authors
Thank you for your revision. However, please include the exact concentration of each compound mentioned in the abstract. Phrases like "rich in minerals" are too broad. Also, clarify what you mean by "1 g/mL" as it is ambiguous. Furthermore, the abstract is too short and should be expanded for better clarity and detail.
In addition, kindly include the chromatogram as supplementary data. Validation of GC, UHPLC, and LC-MS methods should also be provided in the supplementary materials. This could include the limit of detection (LOD), limit of quantification (LOQ), and the calibration standard curve.
Comments on the Quality of English LanguageFine, could be improved.
Author Response
Thank you for your revision. However, please
- include the exact concentration of each compound mentioned in the abstract. Phrases like "rich in minerals" are too broad.
A: Thanks for your comments.
As your suggestions, we added the details of the phrases in the abstract. All the changes were marked with blue.
- Also, clarify what you mean by "1 g/mL" as it is ambiguous.
A: Thank you for your comments.
“1 g/mL” means the concentration of the extracts. To make the expression clearer, we changed the sentence to “The leaf extracts at a concentration of 1mg/L demonstrated significant inhibitory effects on α-glucosidase (84.17%) and pancreatic lipase (77.54%)”. We marked the changes with blue in the abstract part.
- Furthermore, the abstract is too short and should be expanded for better clarity and detail.
A: Thank you for your suggestion. We have made revisions to the abstract section as request.
- In addition, kindly include the chromatogram as supplementary data.
A: Thanks for your comments.
We added the HPLC chromatogram as Figure S4 and S5 in supplementary materials.
- Validation of GC, UHPLC, and LC-MS methods should also be provided in the supplementary materials. This could include the limit of detection (LOD), limit of quantification (LOQ), and the calibration standard curve.
A: Thanks for your comments.
The LOD was calculated and the details were presented in Table S3 in supplementary materials.